# The Association Between Preoperative Sarcopenia and Sarcopenic Obesity and the Occurrence of Postoperative Complications in Patients Undergoing Pancreaticoduodenectomy for Periampullary Malignancies—A Literature Review

**DOI:** 10.3390/nu16203569

**Published:** 2024-10-21

**Authors:** Jakub Ciesielka, Krzysztof Jakimów, Karolina Majewska, Sławomir Mrowiec, Beata Jabłońska

**Affiliations:** 1Student’s Scientific Association, Department of Digestive Tract Surgery, Faculty of Medical Sciences in Katowice, Medical University of Silesia, 40-752 Katowice, Poland; krzysztof.jakimow@gmail.com; 2Department of Digestive Tract Surgery, Faculty of Medical Sciences in Katowice, Medical University of Silesia, 40-752 Katowice, Poland; majewskakarolina1008@gmail.com (K.M.); mrowasm@poczta.onet.pl (S.M.)

**Keywords:** pancreaticoduodenectomy, sarcopenia, sarcopenic obesity, pancreatic ductal carcinoma, biliary tract neoplasms

## Abstract

Background: Sarcopenia and sarcopenic obesity, perceived as a reflection of cancer-induced cachexia, are often diagnosed in patients with periampullary malignancies. The pathophysiology of those conditions is multifactorial regarding the tumor microenvironment, immunological response, and the relationship to surrounding tissues. Methods: The PubMed and SCOPUS databases were systematically searched between November 2023 and December 2023. A total of 254 studies were primarily identified. Regarding the inclusion and exclusion criteria, 26 studies were finally included in the review. Results: Evaluated papers disclosed that sarcopenia was significantly associated with a higher incidence of postoperative complications, including pancreatic fistula (POPF) type B and C, with the odds ratio (OR) ranging from 2.65 (95%CI 1.43–4.93, *p* = 0.002) to 4.30 (95%CI 1.15–16.01, *p* < 0.03). Sarcopenic patients also suffered more often from delayed gastric emptying (DGE) with an OR of 6.04 (95%CI 1.13–32.32, *p* = 0.036). Infectious complications, postoperative hemorrhage, and intra-abdominal abscesses occurred more often in sarcopenic patients. Surgical complications were also noted more frequently when sarcopenic obesity was present. Preoperative nutritional prehabilitation seems to reduce the risk of postoperative complications. However, more prospective studies are needed. Conclusions: Sarcopenia and sarcopenic obesity were associated with a higher incidence of multiple postoperative complications, including POPF (type B and C), DGE, hemorrhage, and infectious complications.

## 1. Introduction

Sarcopenia, defined as the progressive and generalized decrease of skeletal muscle mass, is often confirmed in patients with various periampullary malignancies [1]. It is estimated that sarcopenia occurs in up to 70% of patients diagnosed with pancreatic ductal adenocarcinoma (PDAC) [2]. This state is widely recognized as a reflection of frailty syndrome and the sensitive predictive marker of poorer survival in patients with various gastrointestinal malignancies [3,4,5]. Sarcopenic obesity, which is defined as the prevalence of sarcopenia and increased body mass index (BMI) values, has also been found in patients with PDAC with a frequency of 36.6% [6]. Similarly to sarcopenia, sarcopenic obesity is independently correlated with poorer survival in patients with various digestive tract malignancies [7,8,9]. However, the relationship between sarcopenia and sarcopenic obesity and increased risk of postoperative complications in individuals who underwent pancreaticoduodenectomy (PD) for periampullary malignancies remains vague, with the results differing among the studies. Prehabilitation in the patients with PDAC was believed to have a positive impact on the outcomes; however, there is a lack of multicenter studies that would support this view.

The aim of this review is to assess the impact of sarcopenia and sarcopenic obesity on postoperative outcomes in patients with periampullary tumors who underwent PD. This paper will discuss if there is a correlation between sarcopenia or sarcopenic obesity and postoperative complications in patients undergoing PD for PDAC.

## 2. The Definition of Sarcopenia and Sarcopenic Obesity

There is no standardized definition of sarcopenia, and those existing differ among the variable guidelines. The first definition of sarcopenia was stated by Baumgartner et al. [10], in which, according to the authors, sarcopenia is defined as the muscle mass index of two or more standard deviations below reference values (≤−2 SD) in young and healthy individuals. Regarding the European Working Group of Sarcopenia in Older People (EWGSOP) guidelines, sarcopenia is characterized as a generalized loss of muscle mass and strength, burdened by extensive physical disability and poor quality of life leading to death [11]. According to the International Working Group of Sarcopenia (IWGS) and the European Society for Clinical Nutrition and Metabolism Special Interest Groups (ESPEN-SIG), sarcopenia is defined as a condition of low skeletal muscle and function [12,13]. Notably, according to these guidelines, sarcopenia can occur with decreased muscle mass alone or in connection with increased BMI index, resulting in sarcopenic obesity [12].

## 3. The Pathophysiology of Sarcopenia and Sarcopenic Obesity in Patients with Periampullary Malignancies

Sarcopenia in patients with various periampullary malignancies is caused by both molecular and mechanical alterations during the course of the disease [14]. The detailed pathogenesis of sarcopenia and sarcopenic obesity has been widely evaluated in patients with PDAC. It is worth noting that sarcopenia in PDAC-burdened patients, is not only the result of decreased food intake, during the disease, but a multifactorial reflection of a lethal disease. The mechanisms leading to sarcopenia in patients with PDAC involve the extensive systemic inflammation connected with the tumor microenvironment, immunological response, and the relationship between the tumor and surrounding tissues [14].

It has been proven that increased serum concentrations of inflammatory cytokines induced by the tumor, including interleukin-1 (IL-1), interleukin-6 (IL-6), interleukin-8, tumor necrosing factor alpha (TNF-α), tumor growth factor beta (TGF-β), and C-reactive protein (CRP), are strongly related to PDAC-induced cachexia and sarcopenia [14,15]. Some studies revealed that an increased concentration of IL-1, which occurs in the course of PDAC, can stimulate the hypothalamus to release serotonin, leading to activation of the pro-opiomelanocortin/cocaine–amphetamine-regulated transcript anorexigenic pathway in the hypothalamus. It results in decreased appetite and food intake. Decreased food intake leads to a progressive caloric deficit and results in sarcopenia [16]. Similarly, proinflammatory factors can also inhibit the neuropeptide Y orexigenic pathway in the hypothalamus, contributing to decreased appetite [14]. In addition to the effects of tumor-induced mediators on the central nervous system, sarcopenia during PDAC may also be attributed to the peripheral nervous system. Neural invasion is the relevant biological feature of PDAC, with occurrence ranging from 70% to 95% of cases during the disease course [17]. According to some authors, this state may lead to the activation of astroglial cells in the spinal cord, which may result in the stimulation of the sympathetic nervous system and result in increased lipolysis and myolysis [18].

Considering another component of sarcopenia in patients with PDAC, it has been proven that increased TNF-α and IL-1 concentrations lead to excessive transcription of leptin mRNA, resulting in higher serum leptin concentrations in patients with PDAC and decreased appetite. However, the direct association between hyperleptinemia and progressive weight loss in patients with PDAC was not observed [19].

TGF-β also plays a significant role in the development of sarcopenia. It has been found that TGF-β leads to the activation of mothers against decapentaplegic homologs 2 and 3 proteins that promote extensive muscle loss. It has also been disclosed that TNF-α can mediate muscle protein degradation through the Janus kinase 2/signal transducer and activator of transcription 3 and nuclear factor kappa B pathways [14].

Molecular alterations occurring in patients with PDAC also include the extensive release of tumor-derived factors, including lipid mobilizing factors and protein-induced factors favoring lipo- and proteolysis [20,21]. The first one is considered to interact with B3-adrenergic receptors, leading to lipolysis by activating adenylate cyclase. It has also been proven that protein-induced factors may contribute to the autophosphorylation of RNA-dependent protein kinase, resulting in the inhibition of translation [20].

Some of the mechanical factors related to PDAC also need to be considered. During the extensive growth of the tumor, the obstruction of the intestines may occur even in up to 13% of patients, leading to pain, nausea, nonbilious vomiting, fatigue, and gastroparesis. These conditions result in decreased food intake and significantly contribute to muscle weight loss [22].

Obesity in patients with PDAC is strongly connected with the inflammation of adipose tissue, resulting in the secretion of proinflammatory cytokines, including TNF-α, TGF-β, IL-6, C-C chemokine receptor type 2 and 5, and monocyte chemoattractant protein-1. These factors are significantly associated with the promotion of sarcopenia in PDAC obese patients [23]. It has also been disclosed that proinflammatory cytokines released from fat tissue during obesity may affect the proliferation potential in muscle satellite cells, leading to decreased muscle regeneration and lower muscle mass [24]. It is worth noting that permanent inflammation during obesity may lead to insulin resistance and the downregulation of glucose transporter protein type-4, impairing glucose uptake and resulting in low skeletal muscle [25].

As the data indicate, sarcopenia and sarcopenic obesity are closely related pathological and multicausal processes, mediated by both the tumor microenvironment, the extensive growth of the tumor with invasion to other tissues, and co-occurring obesity in patients with PDAC. Figure 1 demonstrates the molecular pathophysiology of sarcopenia and sarcopenic obesity in patients with PDAC.

## 4. Imaging of Sarcopenia in Patients with Periampullary Malignancies

Computed tomography (CT) constitutes the gold standard diagnostic method for estimating whole-body muscle mass, making it the most reliable tool for diagnosing sarcopenia [26]. CT enables assessment of the skeletal muscle area (SMA) as well as muscle attenuation, enabling the determination of muscle composition. To calculate the SMA, the appropriate region of interest is defined using cross-sectional CT scans with dedicated software [27]. The abdominal muscle area is the most common muscle region where CT scans are acquired, especially at the level of the third lumbar vertebrae [28,29,30]. The SMA concerning the abdominal musculature consists of psoas, para-spinous, transversus, and rectus abdominis, as well as internal and external oblique muscles [31]. To calculate standardized, height-adjusted skeletal muscle index (SMI), the SMA is divided by the patient’s height squared as stated in the following equation:$$SMI=\frac{SMA\left[cm^{2}\right]}{height\left[m^{2}\right]}$$

The sex-adjusted SMI values determining sarcopenia can vary among different studies. However, sarcopenia is generally diagnosed with SMI values below 38.5 cm^2^/m^2^ for women and 52.4 cm^2^/m^2^ for men in abdominal muscle area measurements [29,30]. Another approach is to estimate the whole-body skeletal muscle mass based on the cross-sectional total psoas area (TPA), performed at the level of two pedicles of the third lumbar vertebra [32]. To calculate the standardized, body surface-adjusted total psoas area index (TPAI), the TPA is divided by the patient’s height squared according to the following equation:$$TPAI=\frac{TPA\left[cm^{2}\right]}{height\left[m^{2}\right]}$$

The sex-adjusted TPAI values determining sarcopenia vary among different studies; however, according to the consensus, sarcopenia is generally diagnosed with TPAI values < 385 mm^2^/m^2^ in females and <545 mm^2^/m^2^ in males [33]. Other estimations of whole-body muscle area can be performed concerning the thigh, upper extremity, dorsal, and chest wall muscles [34].

Bioelectrical impedance analysis is another non-invasive and objective method of determining skeletal muscle mass and assessing sarcopenia [35]. However, trivial factors, including altered hydration of patients, may confound the measurements, leading to inaccurate estimations [36]. Magnetic resonance imaging remains another accurate method for assessing sarcopenia, allowing the differentiation of soft tissues and the detection of alterations in muscle composition [37]. Nonetheless, it is expensive and is not routinely performed in patients with PDAC. Dual X-ray absorptiometry is a widely used method to determine whole-body composition, enabling the estimation of lean mass and fat mass [38]. Nevertheless, measurements of muscle mass using this method may be overestimated due to the accumulation of extracellular fluid [27].

Sarcopenia risk analysis with a CT is crucial, especially in patients undergoing PD for periampullary malignancies. In these individuals, the stratification of the low skeletal muscle may be based on the preoperative CT scans, eliminating the need for additional measurements. Other methods are not as cost-effective or may provide false results depending on the hydration level. Moreover, CT is the only routinely performed imaging method. The detailed protocol for muscle mass analysis with CT was described for each study included in this review.

## 5. Methodology

For the purpose of this review, the PubMed and SCOPUS databases were systematically searched between November 2023 and December 2023. The following keywords were used during searching: (“pancreatoduodenectomy” or “pancreaticoduodenectomy”) AND (“sarcopenia” or “low skeletal muscle” or “sarcopenic obesity”). Primarily, 254 studies were identified. After the removal of duplicates (n = 149), the 105 studies were independently screened on titles and abstracts by two authors (J.C., K.J.). The free full text, population-based, cross-sectional as well as cohort studies were systematically evaluated. Case reports and reviews were excluded as only original papers were meant to be included. Studies were regarded as eligible if they included patients with confirmed periampullary malignancies and the co-occurrence of sarcopenia or sarcopenic obesity who were treated by PD. Papers which described other surgical methods were not included to remain consistent. The sarcopenia and sarcopenic obesity had to be assessed with the CT, according to the standardized protocols described in the literature [29,31,39]. Studies in which the authors assessed PD and other surgical approaches, including distal pancreatectomy or total pancreatectomy, were excluded. The discrepancies were solved by consensus between the authors. Finally, 26 studies were included in the review. Figure 2 illustrates the flowchart for the study selection.

## 6. Literature Review

For the purpose of this review, we selected and systematically analyzed 26 studies from the PubMed and SCOPUS databases regarding the inclusion and exclusion criteria.

Tazeoglu et al. [40] in their study assessed the effects of sarcopenia on postoperative complications in patients undergoing PD for PADC. A total of 333 consecutive patients were enrolled in the analysis. The sarcopenia was determined by the psoas muscle area index (PMI), which was assessed by the CT at the level of the third lumbar vertebra. The average from the right and the left psoas muscle area (PMA) was calculated and then divided by the patient’s height squared. The sex-adjusted cut-off for PMI was ≤5.3 for males and ≤3.6 for females. In the sarcopenic group, the authors revealed a significantly higher occurrence of overall surgical complications (65.1% vs. 38.5%, *p* < 0.001), surgical site infection (SSI, 10.8% vs. 4.3%, *p* = 0.022), and biliary fistula (13.3% vs. 3.3%, *p* = 0.029). Multivariate analysis disclosed that sarcopenia was an independent predictor of shorter disease-free survival (HR 2.59, 95%CI 1.79–3.73, *p* < 0.001) and increased mortality (HR 5.67, 95%CI 3.58–8.98, *p* < 0.001), compared to the non-sarcopenic patients.

In another study, 202 patients with PADC and periampullary malignancies were assessed by Pecorelli et al. [29] to disclose the impact of sarcopenia and sarcopenic obesity on postoperative complications. Cut-off values of height-normalized total abdominal muscle area (TAMA) were set at 52.4 cm^2^/m^2^ for men and 38.5 cm^2^/m^2^ for women. According to these cutoffs, sarcopenia has been found in 65.3% of patients. Using visceral fat area (VFA), the VFA/TAMA ratio was calculated for each patient. The authors did not note the difference in survival in the 30-day (3.0% vs. 3.8%, *p* = 1.0) and 60-day periods (3% vs. 7.6%, *p* = 0.224) for patients without sarcopenia to those with it. The authors reveal a similar frequency of pancreatic fistula (26.0% vs. 22.7%, *p* = 0.729) and hemorrhage (13.0% vs. 7.6%, *p* = 0.221) in patients with sarcopenia compared to non-sarcopenic patients. There was also no difference in the prevalence of cardiorespiratory complications (15.2% vs. 7%, *p* = 0.119), delayed gastric emptying (DGE, 12.1% vs. 11%, *p* = 0.885), wound infection (9.8% vs. 14%, *p* = 0.345), bile leak (6.8% vs. 7%, *p* = 1.000), and reoperation rate (9.1% vs. 9.0%, *p* = 0.902) in sarcopenic patients compared to individuals with normal muscle mass. Patients in both groups were hospitalized for similar periods of time [12 (IQR 9–17) vs. 12 (IQR 9–16) days, *p* = 0.785]. What is interesting is that when patients suffered from major complications, the VFA/TAMA ratio > 3.2 was a significant predictor of increased mortality (OR 6.33, 95%CI 1.37–29.21, *p* = 0.018).

In a later study, Pecorelli et al. [41] tried to evaluate the impact of sarcopenic obesity on failure to rescue (FTR) from major surgical complications among 938 patients who underwent PD due to PDAC, periampullary cancer, or neuroendocrine cancer of the pancreas. TAMA was a determinant of sarcopenia, while the VFA/TAMA ratio > 3.2 was the marker of sarcopenic obesity. The difference in rates of FTR and rescued patients was not relevant (60.9% vs. 54.6%, *p* = 0.646). FTR patients had a significantly higher VFA/TAMA ratio (4.1 vs. 3.0, *p* = 0.009), with a value of 3.2 as the best predictor of FTR (78.3% vs. 46.4%, *p* = 0.010). Sarcopenic obesity was associated with FTR on multivariable analysis (OR 5.71, 95%CI 1.58–20.72, *p* = 0.008). Those patients were also more prone to be FTR if postoperative complications occurred: postoperative pancreatic fistula (POPF) type B and C (43% vs. 12%, *p* = 0.009), hemorrhage (36% vs. 5%, *p* = 0.021), reoperation (45% vs. 14%, *p* = 0.040), and patients admitted to the intensive care unit (ICU, 38% vs. 7%, *p* = 0.033).

In the study conducted by Namm et al. [42], the authors tried to measure the influence of TPAI on postoperative outcomes in patients with PDAC. In total, 116 patients were included in the study. There was a significantly higher risk of SSI or pancreatic fistula in patients with high TPAI (OR 3.12, *p* = 0.019). The prevalence of sarcopenia compared to normal muscle mass was not correlated with major complications (*p* = 0.93), readmission to ICU (*p* = 0.19), or readmission to a hospital within 90 days (*p* = 0.13). TPAI was of predictive value for SSI, including POPF type B and C (assessed manually: OR 3.50, *p* = 0.034; semi-automated assessment: OR 4.23, *p* = 0.014).

Stretch et al. [43] in their study assessed the effect of sarcopenia on postoperative surgical results in patients undergoing PD for pancreatic and periampullary adenocarcinomas. Sarcopenia was evaluated by computed tomography at the body of the third lumbar vertebrae and expressed as SMI. The cut-off values for SMI were <47.7 cm^2^/m^2^ for males and <36.5 cm^2^/m^2^ for females. Among 123 patients, sarcopenia was identified in 40.7% of patients. There was no difference between sarcopenic and non-sarcopenic patients when blood loss, operative time, and ICU length of stay were considered. Complications such as DGE, anastomotic leak, and infectious complications did not differ relevantly. Patients with sarcopenia presented shorter mOS (26.4 vs. 16.0 months, *p* = 0.005), compared to non-sarcopenic patients.

In a retrospective study supervised by Tankel et al. [44], 61 patients after PD for PDAC and other types of pancreatic neoplasms were assessed to determine the impact of sarcopenia on the prevalence of postoperative complications. The TPAI threshold for sarcopenia was 83.41 cm^2^/m^2^ for males and 65.28 cm^2^/m^2^ for females. A total of 26.2% of patients met those criteria. Sarcopenic and non-sarcopenic patients did not differ in the following categories: intraoperative blood loss (150 vs. 50 mL, *p* = 0.780), duration of surgery (4.5 hrs. vs. 4.3 hrs., *p* = 0.298), Clavien–Dindo complications >III (5 vs. 11, *p* = 0.584), occurrence of the POPF type B and C (4.9% vs. 19.7, *p* = 0.738), and length of hospital stay (11.0 vs. 13.0 days, *p* = 0.735). However, DGE was more common in the first group (43.8% vs. 17.8%, *p* = 0.045) with an OR of 6.042 (95%CI 1.131–32.319, *p* = 0.036).

Jang et al. [28] in their study tried to assess whether sarcopenia or visceral obesity can determine the occurrence of POPF type B and C. Sarcopenia was defined as a TAMA index ≤ 52.4 cm^2^/m^2^ in males and ≤38.5 cm^2^/m^2^ in females. Sarcopenic obesity was stated as VFA/TAMA > 3.2. Of 284 patients treated with PD, 67.3% were sarcopenic and 29.6% had sarcopenic obesity. The occurrence of POPF and major postoperative complications did not differ in patients with sarcopenia (*p* = 0.751 and *p* = 0.958, respectively). In total, 48.1% of patients with POPF had VFA/TAMA exceeding 3.2, which reached a significant difference (*p* = 0.001) in this group. In the univariate logistic regression analysis, sarcopenic obesity was predictive of POPF and it was the only significant predictor in the multivariate analysis (OR 2.65, 95%CI 1.43–4.93, *p* = 0.002).

The aim of the study provided by Linder et al. [45] was to assess the morbidity after PD in 139 patients with periampullary carcinomas. Sarcopenia was defined by two factors: SMI (<41 cm^2^/m^2^ for females, <43 cm^2^/m^2^ for males with BMI < 25 kg/m^2^, and <53 cm^2^/m^2^ for males with BMI ≥ 25 kg/m^2^) and skeletal muscle attenuation measured in Hounsfield Units (HU, <41 HU for both sexes with BMI < 25 kg/m^2^ and <33 HU for both sexes with BMI ≥ 25 kg/m^2^). A total of 60 patients were classified as sarcopenic using both values. Sarcopenia was associated with a significant risk of POPF (type B and C, OR 4.30, 95%CI 1.154–16.005, *p* < 0.03).

Shintakuya et al. [46] conducted a study that included 112 patients who underwent PD. Sarcopenia was defined using PMI, and the lowest quartiles were sarcopenic. In total, 33 patients were described as sarcopenic. When DGE was considered, sarcopenia was a significant risk factor for that complication (58% vs. 18%, *p* = 0.002).

A retrospective study was provided by Ratnayake et al. [47] to assess the influence of sarcopenic obesity on the outcomes among 89 patients who underwent PD for PDAC. Sarcopenia was assessed using three approaches: SMI (<43 cm^2^/m^2^ in men with a BMI <25 kg/m^2^, <53 cm^2^/m^2^ in men with a BMI ≥ 25 kg/m^2^, and <41 cm^2^/m^2^ in women), psoas muscle index (<5.9 cm^2^/m^2^ in men and <4.1 cm^2^/m^2^ in women), and skeletal muscle attenuation (<33.9 HU in men and <30.9 HU in women). Sarcopenic obesity was a condition when sarcopenia was present with a BMI over 25 kg/m^2^. In total, 49 patients were classified as sarcopenic using PMI, 44 patients using SMI, and 25 patients using skeletal muscle attenuation. Only by using the last method the duration of surgery was significantly longer in non-sarcopenic patients (*p* = 0.014). Blood loss volume and the rates of POPF type B and C, DGE, surgical site infection, hemorrhage, and overall postoperative morbidity did not differ between patients with and without sarcopenia. The same situation was observed regarding the duration of ICU hospitalization, ICU readmissions, reoperation rates, cardiac, respiratory, and renal complications as well as the length of in-hospital hospitalization. Although, the only factor of overall postoperative morbidity (OR 1.241, *p* = 0.041) was sarcopenic obesity.

Ryu et al. [48] presented a study that included 548 patients who underwent PD for pancreatic cancer. In the paper, sarcopenia was defined as SMI < 50.18 cm^2^/m^2^ for men and SMI < 38.63 cm^2^/m^2^ for women, while the cut-off value for sarcopenic obesity was proposed at VFA/SMI 2.5 m^2^. Preoperative sarcopenia was stated in 252 patients (57.1% were men, 29.7% were women), while sarcopenic obesity occurred in 202 of them. Greater loss of blood during operation was observed in patients with sarcopenic obesity (660.2 mL vs. 537.1 mL, *p* = 0.023). In the sarcopenic group, compared to individuals with normal muscle mass, the incidence of several adverse events, including major complications (Clavien–Dindo classification ≥ 3, 22% vs. 16.3%, *p* = 0.093), DGE (7.4% vs. 3.6%, *p* = 0.051), post-PD hemorrhage (3.4% vs. 3.2%, *p* = 0.894), wound infection (8.4% vs. 6.7%, *p* = 0.456), and bile leak (0.7% vs. 0%, *p* = 0.502) was higher, however did not differ among the compared groups. When sarcopenic obesity and non-sarcopenic obesity groups were considered, all conditions were also statistically insignificant except for POPF type B and C, which was observed more frequently in patients with sarcopenic obesity (*p* < 0.001). The 5-year survival rate for sarcopenia-burdened patients was notably worse (23.4% vs. 28.4%, *p* = 0.046), but this difference was not significant when sarcopenic obesity and non-sarcopenic obesity were considered (28.5% vs. 24.8%, *p* = 0.877).

In a retrospective study provided by Aoki et al. [49], 180 patients who underwent PD for pancreatic cancer were involved to assess the potential influence of sarcopenia on the incidence of postoperative complications. Sarcopenia was assessed by SMI, grip strength, and gait speed. SMI was calculated as the appendicular lean body mass divided by the square of height. The SMI cut-off points for sarcopenia were stated as <7 kg/m^2^ in men and <6 kg/m^2^ in women. The cut-off points for grip strength were <27 kg in men and <16 kg in women, where the cut-off point for gait speed was ≤0.8 m/s in both sexes. Patients with low SMI and low grip strength values were considered sarcopenic. The authors revealed no associations between sarcopenia and the incidence of postoperative complications, including POPF type A, B, and C, DGE, wound infection, intra-abdominal abscess, as well as bacteriemia and pneumonia. Notwithstanding, in a multivariate logistic analysis, sarcopenia was identified as an independent risk factor for a significantly lower recurrence-free rate (HR 4.48, 95%CI, 1.68–11.98, *p* = 0.003) and poorer overall survival (HR 3.25, 95%CI, 1.19–8.86, *p* = 0.021).

Pessia et al. [30] retrospectively identified 76 consecutive patients who underwent PD for PDAC. SMA was measured by computed tomography at the body of the third lumbar vertebra, normalized for body length, and expressed as SMI. The cut-off points for sarcopenia were defined as SMI ≤ 52.4 cm^2^/m^2^ for men and ≤38.5 cm^2^/m^2^ for women. The authors revealed a higher duration of hospitalization in sarcopenic patients compared to non-sarcopenic individuals (21 days vs. 17 days). A relevantly lower recurrence-free survival, as well as overall survival, have been found in sarcopenia-burdened patients compared to individuals with normal SMI.

Centonze et al. [50] retrospectively analyzed 110 cases of patients who underwent PD for PDAC, distal bile duct cancer (DBDC), adenocarcinoma (AC), intraductal papillary mucinous neoplasm (IPMN), pancreatic neuroendocrine tumor (PNET), gastrointestinal stromal tumor, or chronic pancreatitis to assess the potential impact of sarcopenia on postoperative outcomes. To define sarcopenia, the PMA was measured by computed tomography and expressed as a Hounsfield unit average calculation (HUAC). The HUAC was calculated using the following equation:$$HUAC=HounsfieldUnit\times PMA+Unit\times \frac{\;}{TotalPMA\times 2}$$

The cut-off points for defining sarcopenia were stated as a HUAC < 37 HU for males and <14.21 HU for females. Considering the above-mentioned criteria, sarcopenia has been found in 32.7% of patients. The authors revealed a significantly higher rate of POPF type C in sarcopenic patients compared to non-sarcopenic individuals (50% vs. 11.4%, *p* = 0.005). However, no association between sarcopenia and DGE, biliary fistula, surgical site infection, postoperative hemorrhage, and sepsis has been found.

Sur et al. [51] disclosed a negative impact of sarcopenia on the postoperative outcome in patients undergoing PD for PDAC. In the study, 104 past medical records were retrospectively analyzed. TPAI and density were measured by computed tomography at the third lumbar vertebra and expressed in cm^2^/m^2^ and weighted average HU, respectively. The authors revealed that low HU values were independently associated with serious surgical complications according to the American College of Surgeons National Surgical Quality Improvement Program (NSQIP) (r = −0.31, *p* = 0.0098), as well as a grade ≥ 3 of Clavien–Dindo complications (r = −0.29, *p* = 0.0183).

A study performed by Takagi et al. [52] involved 291 patients who underwent PD for periampullary tumors, including PDAC, DBDC, AC, duodenal carcinoma, IPMN, and others. Sarcopenia was defined by skeletal body index, calculated by dividing the cross-sectional skeletal muscle area (SBA) at the level of the third lumbar vertebrae by BMI. The cut-off values for defining sarcopenia were <68.5 cm^2^/m^2^ for men and <52.5 cm^2^/m^2^ for women. Preoperative sarcopenia was diagnosed in 25.1% of individuals. Patients with sarcopenia, compared to non-sarcopenic individuals, presented a significantly higher rate of infectious complications (67.3% vs. 40.2%, *p* < 0.001), abdominal fluid collections (43% vs. 25%, *p* = 0.009), bacteriemia (9.1% vs. 1.2%, *p* = 0.009), pneumonia (5.5% vs. 0.6%, *p* = 0.02), and intra-abdominal abscess (10.9% vs. 3.7%, *p* = 0.04). The authors did not reveal a significant correlation between sarcopenia and adverse events, including major postoperative complications, POPF type B and C, DGE, wound infection, catheter-related infection, cholangitis, enteritis, and duration of hospitalization. The 5-year survival rates in the sarcopenia cohort were significantly lower compared to non-sarcopenic individuals (23.4% vs. 28.4% *p* = 0.046).

In the study performed by Sui et al. [53], 150 patients who underwent PD for PDAC, IPMN, PNET, ampullary carcinoma, duodenal carcinoma, and DBDC were included. Sarcopenia was defined as SMI < 40.5 cm^2^/m^2^ for men and <33.5 cm^2^/m^2^ for women. The incidence of postoperative complications, including DGE, bile leakage, intra-abdominal abscess, as well as the duration of hospitalization, did not differ significantly; however, in sarcopenic patients, a significantly lower incidence of POPF type B and C was observed (19.5% vs. 31.6%, *p* = 0.04). Five-year OS in patients with sarcopenia was significantly lower compared to non-sarcopenic patients (51.6% vs. 57%, *p* = 0.009).

In the study provided by Nauheim et al. [54], sarcopenia was stated as the lowest quartile regarding PMA values. Sarcopenia was associated with postoperative complications (26.8% vs. 8.5%, *p* < 0.001) and grade 2 complications according to the Clavien–Dindo classification. The frequency of complications, including gastro/duodenostomy leak, any grade of pancreatic fistula, as well as intra-abdominal abscess and intra-abdominal bleeding, was not significantly higher in patients with low PMA values compared to normal. The authors have proven that low PMA was independently associated with prolonged hospitalization compared to the normal PMA cohort. The percentage of 30- and 90-day mortality did not differ between the low and normal PMA groups.

A study conducted by Nishida et al. [55] included 266 patients who underwent PD for PDAC. Sarcopenia was stated as SMI < 43 cm^2^/m^2^ + BMI < 25 kg/m^2^ or SMI < 53 cm^2^/m^2^ + BMI of ≥25 kg/m^2^ in men, and SMI < 41 cm^2^/m^2^ in women. Patients with sarcopenia compared to non-sarcopenic patients experienced significantly higher rates of major surgical complications, stated as Clavien–Dindo > 3 (34.1% vs. 15.7%, *p* = 0.001) as well as POFP type ≥ B (22.0% vs. 10.4%, *p* = 0.011). Reoperation rates and duration of hospitalization did not differ among the compared groups.

Sandini et al. [56] tried to determine the role of sarcopenic obesity on the occurrence of complications after surgical management. Thus, 124 patients after PD due to pancreatic cancer were retrospectively analyzed. The cut-off value for the TAMA was set at <41 cm^2^/m^2^ for females and of TAMA < 43 cm^2^/m^2^ (with BMI < 25 kg/m^2^) or <53 cm^2^/m^2^ (with BMI ≥ 25 kg/m^2^) for males to determine sarcopenia. This condition was observed then in 24.2% of patients. Adverse effects after surgery (POPF type A, B, and C, biliary leak, DGE, sepsis, hemorrhage, abscess, and wound infection) did not differ relevantly between sarcopenic and non-sarcopenic groups. However, if sarcopenic obesity was stated, the occurrence of abscesses was higher (43.3 vs. 20.6, *p* = 0.007). One-third of all patients suffered from major complications (Clavien–Dindo ≥ III), but only the VFA/1 ratio was associated with them (OR 3.20, 95%CI 1.35–7.60, *p* = 0.008) in the multivariate logistic regression. The mortality rate was not significantly different between the groups (4.3% in the non-sarcopenic group vs. 6.9% in the sarcopenic group, *p* = 0.564).

Yoo et al. [57] conducted a retrospective study among 257 PDAC patients to determine if body composition measurements can predict the occurrence of pancreatic fistula and OS after PD. The measurements were acquired using convolutional neural networks trained to assess CT images for SMA, subcutaneous adipose tissue, and visceral adipose tissue. Those values were then normalized to height square with the cut-off for low skeletal muscle set at 49.5 cm^2^/m^2^. SMI was not found to have a predictive value for POPF grade B or C (*p* = 0.211). However, it can predict the 1-, 3-, and 5-year OS rates when considering high and low skeletal muscle levels [86.2% vs. 71.9%, 39.7% vs. 30.3 %, and 28.8% vs. 16.3%, respectively (*p* = 0.026)].

To assess the potential impact of sarcopenia on the occurrence of postoperative complications, Cai et al. [58] included 129 consecutive patients who underwent PD for PDAC, DBDC, AC, pancreatic cystic neoplasm, neuroendocrine tumor, and chronic pancreatitis. Sarcopenia was assessed with CT at the third lumbar vertebra and expressed as SMI by dividing TAMA by the patient’s height squared. The sarcopenia cut-off points were stated as SMI < 42.2 cm^2^/m^2^ for men and <33.9 cm^2^/m^2^ for women. The multivariate analysis revealed that sarcopenia was a sensitive predictive marker of the occurrence of both POPF type B and C (OR 2.999, 95%CI: 1.365–6.596, *p* = 0.006) and higher Charlson Comorbidity Index (CCI) scores above 26.2 (OR 3.425, 95%CI: 1.541–7.613, *p* = 0.003).

Menozzi et al. [59] in their study tried to determine the role of preoperative sarcopenia in the development of postoperative complications in individuals undergoing PD, total pancreatectomy, and distal pancreatic resection for PDAC. The measurements of skeletal muscle tissue were performed by CT at the level of the third lumbar vertebra. SMI was expressed by dividing the total lumbar area by the patient’s height squared. The sex-specific cut-off points were stated for men and women as SMI < 52.4 cm^2^/m^2^ and <38.5 cm^2^/m^2^, respectively. Preoperative sarcopenia was observed in 56.3% of patients. The authors revealed an independent relationship between sarcopenia and digestive hemorrhages (OR 0.10, 95%CI 0.01–0.72, *p* = 0.03). Nevertheless, the impact on a higher incidence of POPF type B and C, biliary fistula, DGE, and Clavien–Dindo complications score III–V was not observed.

Xu et al. [60] in their study assessed the impact of sarcopenia on the occurrence of postoperative complications in patients undergoing PD for both malignant and benign entities. In total, 152 consecutive patients were included in the study. PDAC was confirmed in 28.3% of all individuals. Skeletal muscle was measured by CT on the level of the third lumbar vertebra by assessing TPA. TPA was then divided by the patient’s height and expressed as TPAI. Sarcopenia was defined as TPAI values below ≤4.78 cm^2^/m^2^ for males and ≤3.46 cm^2^/m^2^ for females. Sarcopenia was found in 38.8% of all individuals. The percentage of major complications was significantly higher in patients burdened with sarcopenia compared to non-sarcopenic individuals (40.7% vs. 7.5%, *p* < 0.001). The rate of reoperation, as well as the duration of hospitalization, was significantly increased in patients with low skeletal muscle. Patients with sarcopenia also presented a significantly higher percentage of in-hospital mortality compared to non-sarcopenic patients (6.8% vs. 1.1%); however, that association was not significant (*p* = 0.055).

Guarneri et al. [61] assessed 371 patients who underwent PD for PDAC. Sarcopenia cut-offs were established by two factors: SMI (<41 cm^2^/m^2^ for females, <43 cm^2^/m^2^ for males with BMI < 25 kg/m^2^, and <53 cm^2^/m^2^ for males with BMI ≥ 25 kg/m^2^) and skeletal muscle attenuation measured in Hounsfield Units (HU, <41 HU for both sexes with BMI < 25 kg/m^2^ and <33 HU for both sexes with BMI ≥ 25 kg/m^2^). Sarcopenic obesity was present if the criteria stated by Pecorelli et al. [41] were met. In total, 51.4% of the patients included were treated with preoperative chemotherapy; 80.1% of the cohort was described as sarcopenic; and sarcopenic obesity was observed in 33.2% of patients. On multivariable analysis, sarcopenic obesity was associated with major postoperative complications (OR = 1.68, 95%CI 1.02–2.98, *p* = 0.048). POPF types A, B, and C were present more often in the sarcopenic obese group (28.5% vs. 16.9%, *p* = 0.010). There were no differences in the occurrence of DGE, postoperative hemorrhage, and SSI. Sarcopenia and sarcopenic obesity did not have an impact on OS.

In total, 169 patients suffering from PDAC, DBDC, IPMN, pancreatic neuroendocrine tumor, and AC were evaluated by Hayashi et al. [62], in order to determine the correlation between sarcopenic obesity and POPF type B/C. Cut-offs for sarcopenia were established as an SMI of <52.4 cm^2^/m^2^ for men and <38.9 cm^2^/m^2^ for women. A VFA/SMI > 1.58 was the cut-off for sarcopenic obesity. Sarcopenic obesity was more frequently observed when POPF type B and C occurred (71.1 vs. 33.6%, *p* < 0.001). It was also a risk factor for developing POPF (OR 2.94, 95%CI 1.09–7.94, *p* = 0.033).

All papers that were included in this review are presented in Table 1.

## 7. Discussion

The effects of sarcopenia and sarcopenic obesity in patients undergoing PD for periampullary malignancies vary across the assessed studies. We systematically evaluated 26 papers indicating that patients with sarcopenia and sarcopenic obesity are more prone to developing certain adverse effects, including POPF, DGE, digestive hemorrhage, and intra-abdominal abscesses after the surgery. According to the authors, the percentage of longer duration of hospitalization was also significantly higher in sarcopenia-burdened individuals. The most common parameter determining sarcopenia was SMI, which was performed in 13 of 26 studies, followed by TPAI, which was assessed in five studies. The cut-off values determining sarcopenia varied across the compared studies.

Linder et al. [45], Centonze et al. [50], Nishida et al. [55], and Cai et al. [58] presented that sarcopenic patients are more prone to develop POPF type B and C. However, Jang et al. [28] stated that sarcopenia alone was not associated with POPF, contrary to sarcopenic obesity, which was a risk factor for this adverse effect. Similarly, Ryu et al. [48], Guarneri et al. [61], and Hayashi et al. [62] described that patients with sarcopenic obesity presented a significantly higher percentage of POPF type B and C. Across the compared studies, the odds ratio of developing POPF ranged from 2.65 (95%CI 1.43–4.93, *p* = 0.002) to 4.30 (95%CI 1.15–16.01, *p* < 0.03). The authors tried to explain the increased incidence of POPF in patients with PDAC and sarcopenia. According to them, factors that may lead to POPF include the decreased diameter of the pancreatic duct and parenchymal thickness, which result in the leakage of pancreatic–intestinal anastomoses [45,48,55]. This condition may also be explained by the increased visceral adipose tissue that generates an inflammatory response by secreting adipokines (leptin, TNF-α, IL-1, IL-6), which contributes to weaker immunological response and impaired healing after the PD [28,48,58]. Secondly, POPFs may occur due to low blood perfusion in the pancreas because of excessive adipocyte deposition along the pancreatic duct. Those states may lead to perioperative pancreatitis [48]. Guarneri et al. [61] postulated that POPF in sarcopenic obese patients is associated with increased intraoperative blood loss, prolonged surgery, and soft pancreatic stump texture. Interestingly, in eight studies, POPF occurred more frequently in the patients with sarcopenia or sarcopenic obesity but the difference was not significant. This is a controversial finding which was not sufficiently explained by most papers. More studies are needed to assess the correlation correctly.

Unexpectedly, Namm et al. [42] suggested that patients suffering from sarcopenia had a decreased probability of SSI or POPF. This correlation may be justified by the lower amount of water in the patient’s body, decreased exocrine function of the pancreas, and the high proportion of firm glands and large ducts. However, those patients were more likely to be discharged to skilled nursing facilities due to age, BMI, malnutrition, and other comorbidities.

Tankel et al. [44] noted that the difference in the occurrence of DGE was significant, with nearly one-fourth of the sarcopenia-burdened patients suffering from it (OR 6.042, 95%CI 1.131–32.319, *p* = 0.036). A paper provided by Shintakuya et al. [46] disclosed a similar observation regarding this association. This condition may be explained by impaired gastric motility due to malfunctioning of the vagal nerve as well as the intraoperative ischemia of the antrum and the pylorus. That condition could also be caused by the excessive removal of the duodenal tissue during pancreaticoduodenectomy, which is responsible for secreting motilin [44]. However, most papers did not observe this correlation.

Pecorelli et al. [41] also noticed that sarcopenia and especially sarcopenic obesity are risk factors for FTR. According to the authors, it is caused by chronic inflammation, insulin resistance, and frailty, which can make compensation mechanisms insufficient when a postoperative complication occurs. This was the only study that assessed this parameter.

In studies conducted by Tazeoglu et al. [40] and Takagi et al. [52], the authors revealed the connection between sarcopenia and infectious complications in patients undergoing PD for PDAC. This fact may be explained by impaired immune function in patients with sarcopenia and increased levels of adipose tissue. It may lead to increased secretion of adipokines, which contribute to prolonged depletion of the immune system, resulting in infectious complications.

In papers published by Sur et al. [51], Takagi et al. [52], Sandini et al. [56], Menozzi et al. [59], and Xu et al. [60], the authors mentioned that sarcopenia was a factor for a greater number of intra-abdominal abscesses and digestive tract hemorrhages; however, according to Ratnayake et al. [47], only sarcopenic obesity was a factor of postoperative morbidity in patients with those complications. Unfortunately, the authors did not disclose an explanation of why those conditions are more common when sarcopenia is present. We suspect that the formation of intra-abdominal abscesses could be a consequence of the leakage from pancreatic intestinal anastomosis exacerbated by impaired healing in sarcopenic patients. Many studies also focused on the impact of sarcopenia on in-hospital stay and survival after PD. Sarcopenic patients tended to require longer hospitalization [30,54].

According to Ryu et al. [48], Takagi et al. [52], Sui et al. [53], and Yoo et al. [57], the 5-year survival rates were significantly poorer in patients with sarcopenia compared to non-sarcopenic patients. The study conducted by Yoo et al. [57] described impaired 1- and 3-year OS rates (86.2% vs. 71.9% and 39.7% vs. 30.3%, respectively (*p* = 0.026)). Stretch et al. [43] also noted decreased mOS (26.4 vs. 16.0 months, *p* = 0.005). Pessia et al. [30] proved that low skeletal mass is correlated with a lower recurrence-free rate and OS. The same association was found by Aoki et al. [49] (lower recurrence-free rate: HR 4.48, 95%CI 1.68–11.98, *p* = 0.003; OS: HR 3.25, 95%CI 1.19–8.86, *p* = 0.021) despite the difference in postoperative complications being insignificant.

## 8. Future Directions

Regarding the different approaches to measuring sarcopenia and sarcopenic obesity in the aforementioned papers, we believe that it is crucial to develop a universal method of defining both conditions to unify future observations. Considering the discrepancies in the cut-off values, we also postulate that future studies should identify a specific value that will be the most sensitive predictor of the development of postoperative complications. In the future, given the multifactorial nature of sarcopenia, one may wonder about the usefulness of determining pro-inflammatory cytokines in predicting the development of postoperative complications in patients undergoing PD, which could contribute to better perioperative care. However, the usefulness of such interventions requires further research.

Nutritional prehabilitation was introduced as a method that can enhance the outcomes of patients with pancreatic cancer. It can be beneficial for the postsurgical quality of life, nutritional status, and performance in sarcopenia-prone patients [63]. In recent years, several prospective trials in which the authors tried to reduce sarcopenia by multimodal prehabilitation in patients with PDAC have been growing; however, the knowledge about the feasibility of those methods is based on very limited data. Ausania et al. [64] enrolled 40 consecutive patients with confirmed PDAC. Multidimensional prehabilitation, including proper nutrition, pancreatic enzyme replacement therapy, as well as physical and respiratory training, was implemented in 18 individuals. The authors revealed no difference between the experimental and control groups in terms of the number of overall complications, number of major complications (type III–IV), percentage of POPF (type B or C), as well as the duration of hospitalization and percentage of readmissions. Interestingly, the presented study revealed that the percentage of DGE was significantly lower in individuals who underwent multimodal prehabilitation compared to the control group. In the other study conducted by Tsukagoshi et al. [65], the authors enrolled 101 patients with PDAC and preoperatively confirmed sarcopenia, of whom 33 and 30 patients received prehabilitation in terms of nutritional support and rehabilitation therapy, respectively. The authors did not disclose differences between the control and interventional groups in terms of blood loss, duration of postoperative hospitalization, and the percentage of severe postoperative complications (Clavien–Dindo grade ≥ 3). In the presented study, the authors disclosed a significant correlation between the lack of nutritional intervention and a higher risk of clinically relevant POPF (OR 5.57, 95%CI 1.08–28.68, *p* = 0.040). We suggest that prehabilitation in patients with periampullary malignancies and sarcopenia may be beneficial. However, multi-center interventional studies are needed to prove the positive effect on a larger population.

Another interesting possibility is administration of pancrelipase. Matsumoto et al. [66] demonstrated that in sarcopenic patients, it helps to increase the number of cycles of adjuvant chemotherapy (6 vs. 3, *p* = 0.03). The completion rate was also significantly greater when the enzyme was administered (86% vs. 25%, *p* = 0.007) [66]. This method can contribute to a better prognosis for sarcopenic patients with pancreatic cancer.

In addition, preoperative immunomodulating nutrition (IN), using various nutrients, such as glutamine, arginine, omega-3 fatty acids, and nucleotides, should be considered in patients undergoing PD, because the significant influence of IN on the decreased rate of infectious complications and the duration of hospitalization has been proven [67]. According to a 17^th^-strong (grade A) recommendation, preoperatively, oral nutritional supplements should be given to all malnourished cancer and high-risk patients undergoing major abdominal surgery. The 18th recommendation prefers the preoperative (5–7 days before surgery) administration of oral IN supplements, including arginine, omega-3 fatty acids, and nucleotides [67,68,69]. The benefit of IN on postoperative outcome in patients undergoing PD is associated with the fact that IN modulates the inflammatory response and the production of inflammatory mediators as well as favorably modulates systemic and mucosal immunity in patients undergoing PD [67]. The modulation of the immune response in cancer patients undergoing major surgery is very important, because it has been proven that the negative impact of malnutrition on prognosis in cancer patients is related to significant immune system dysfunction in both the cellular and humoral immunity. Therefore, systemic cancer-related inflammatory response and numerous alterations in immune-inflammatory parameters are reported in cancer patients [67,70,71].

## 9. Conclusions

Sarcopenia and sarcopenic obesity are significantly correlated with an increased occurrence of postoperative complications, including POPF type B and C, DGE, infectious complications, intraabdominal abscesses, and postoperative hemorrhage. The reduction of muscle mass was relevantly connected with a longer duration of hospitalization and a higher incidence of ICU admissions. The prehabilitation in patients with periampullary tumors undergoing PD may be beneficial and result in the reduction of some postoperative complications.

## Figures and Tables

**Figure 1 nutrients-16-03569-f001:**
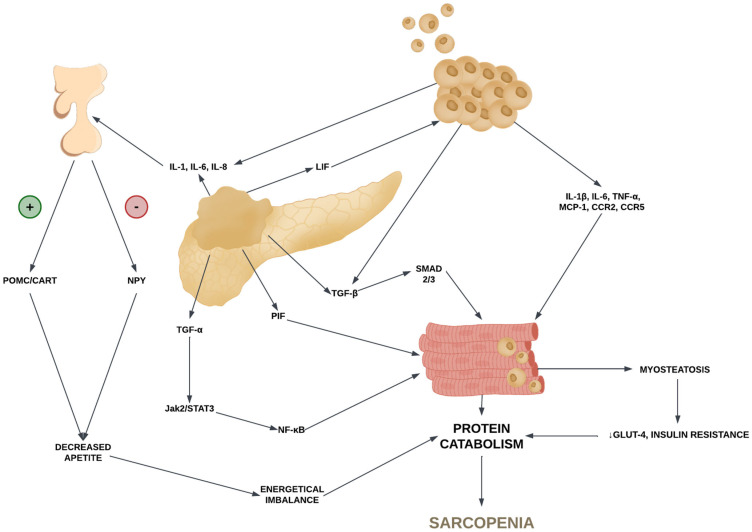
The molecular pathophysiology of sarcopenia and sarcopenic obesity in patients with PDAC. POMC—pro-opiomelanocortin, CART—cocaine–amphetamine-regulated transcript, NPY—neuropeptide Y, IL-1, 1β, 6, 8- interleukin 1, 1β, 6, 8, TGF-α—tumor growth factor alpha, TGF-β—tumor growth factor beta, TNF-α—tumor necrosis factor-alpha, PIF—protein-induced factor, LIF—lipid-included factor; NF-κB—nuclear factor kappa B, CCR2, CCR5—C-C chemokine receptor type 2 and 5, MCP-1—monocyte chemoattractant protein-1, Jak2/STAT3—Janus kinase 2/signal transducer and activator of transcription 3, SMAD2/3—mother against decapentaplegic homologs 2 and 3 proteins, GLUT-4—glucose transporter protein type.

**Figure 2 nutrients-16-03569-f002:**
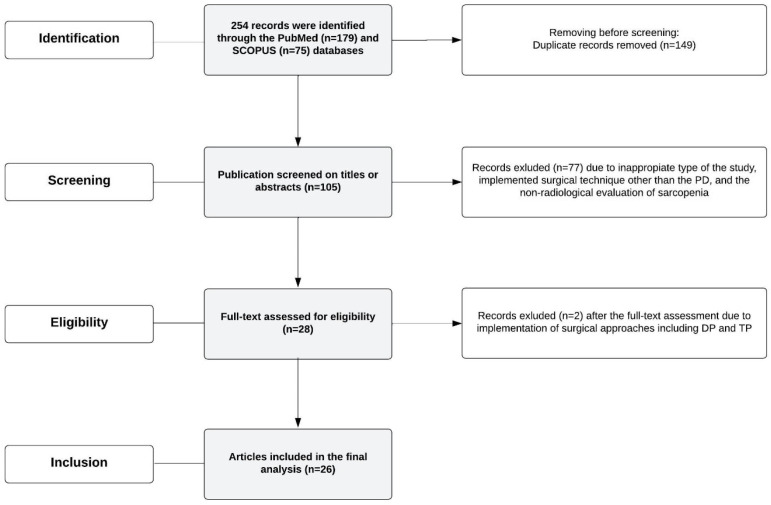
The flowchart for the study selection.

**Table 1 nutrients-16-03569-t001:** Characteristics of studies included in this review. AC—ampullary carcinoma, CCI—Charlson Comorbidity Index, DBDC—distal bile duct cancer, DFS—disease-free survival, DGE—delayed gastric emptying, FTR—failure to rescue, GIST—gastrointestinal stromal tumor, HUAC—Hounsfield unit average calculation, ICU—intensive care unit, IPMN—intraductal papillary mucinous neoplasm, mOS—median overall survival, PMA—psoas muscle area, PMI—psoas muscle index, PNET—pancreatic neuroendocrine tumor, PD—pancreatoduodenectomy (regarded as the Whipple’s procedure), PDAC—pancreatic ductal adenocarcinoma, POPF—postoperative pancreatic fistula, PPPD—pylorus-preserving pancreatoduodenectomy, SBI—skeletal muscle area/body surface area index, SMI—skeletal muscle area, SSI—surgical site infection, SSPPD—subtotal stomach-preserving pancreatoduodenectomy, TAMA—total abdominal muscle area, TPAI—total psoas muscle area index, VFA—visceral fat area.

Ref.	Sample Size	Confirmed Malignancy	Type of Surgery	The Parameter Defining Sarcopenia	Sex-Adjusted Cut-Off Values Defining Sarcopenia (M—Man, F—Female)	The Parameter Defining Sarcopenic Obesity	Sex-Adjusted Cut-Off Values Defining Sarcopenic Obesity (M—Man, F—Female)	Conclusions
Tazeoglu et al. [40]	333	PDAC	PD	TPAI	≤5.3 in M and ≤3.6 in F	-	-	Sarcopenia was correlated with a higher occurrence of overall surgical complications (65.1% vs. 38.5%, *p* < 0.001), SSI (10.8% vs. 4.3%, *p* = 0.022), and biliary fistula (13.3% vs. 3.3%, *p* = 0.029). Low muscle mass was an independent predictor of shorter DFS (HR 2.59, 95%CI 1.79–3.73, *p* < 0.001) and increased mortality (HR 5.67, 95%CI 3.58–8.98, *p* < 0.001).
Pecorelli et al. [29]	202	PDAC, other periampullary tumors	PPPD	TAMA index	<52.4 cm^2^/m^2^ in M, <38.5 cm^2^/m^2^ in F	-	-	There was no difference in the occurrence of POPF type B and C (26.0% vs. 22.7%, *p* = 0.729) and hemorrhage (13.0% vs. 7.6%, *p* = 0.221) among the compared groups. A VFA/TAMA ratio > 3.2 was an independent predictor of increased mortality (OR 6.33, 95%CI 1.37–29.21, *p* = 0.018) in patients with sarcopenia.
Pecorelli et al. [41]	938	PDAC, PNET, other periampullary tumors	PPPD	-	-	VFA/TAMA	>3.2	Sarcopenic obesity was associated with higher FTR in multivariable analysis (OR 5.71, 95%CI 1.58–20.72, *p* = 0.008).
Namm et al. [42]	116	PDAC	PPPD	TPAI	2.75 cm^2^/m^2^	-	-	Sarcopenia was correlated with a higher risk of SSI or POPF type B and C in patients with high TPAI (OR 3.12, *p* = 0.019).
Stretch et al. [43]	123	PDAC	PD	SMI	<47.7 cm^2^/m^2^ in M, <36.5 cm^2^/m^2^ in F	-	-	Patients with sarcopenia presented shorter mOS (26.4 vs. 16.0 months, *p* = 0.005). There was no significant difference in the frequencies of surgical complications between the compared groups.
Tankel et al. [44]	61	PDAC	PD, PPPD	TPAI	83.41 cm^2^/m^2^ in M, 65.28 cm^2^/m^2^ in F	-	-	DGE was more common in the sarcopenic group compared to patients without sarcopenia (OR 6.042, 95%CI 1.131–32.319, *p* = 0.036).
Jang et al. [28]	284	PDAC, AC, DC, DBDC	SSPPD	TAMA index	≤52.4 cm^2^/m^2^ in M, ≤38.5 cm^2^/m^2^ in F	VFA/TAMA	>3.2	Sarcopenic obesity was predictive for POPF type B and C (OR 2.65, 95%CI 1.43–4.93, *p* = 0.002). The occurrence of POPF and major postoperative complications did not differ in patients with sarcopenia.
Linder et al. [45]	139	PDAC, AC, DC, DBDC, pseudotumor after pancreatitis	PD. PPPD	SMI and SMA	SMI: <43 cm^2^/m^2^ in M with BMI < 25 kg/m^2^ and <53 cm^2^/m^2^ in M with BMI ≥ 25 kg/m^2^, <41 cm^2^/m^2^ in F SMA: <41 HU in both sexes with BMI < 25 kg/m^2^ and <33 HU in both sexes with BMI ≥ 25 kg/m^2^	-	-	Sarcopenic patients were associated with an increased risk of POPF type B and C (OR 4.30, CI 1.15–16.01, *p* < 0.03) and the occurrence of adverse effects categorized as grade ≥ 3 in the Clavien–Dindo classification was significantly higher.
Shintakuya et al. [46]	112	PDAC, AC, DC, DBDC, IPMN	PD, PPPD, SSPPD	PMI	32.1 cm^2^/m^2^ in M 24.8 cm^2^/m^2^ in F	-	-	Sarcopenia was correlated with an increased DGE compared to non-sarcopenic patients (58% vs. 18%, *p* = 0.002).
Ratnayake et al. [47]	89	PDAC	PD	SMI, PMI, and SMA	SMI: <43 cm^2^/m^2^ in M with BMI < 25 kg/m^2^ and <53 cm^2^/m^2^ in M with BMI ≥ 25 kg/m^2^, <41 cm^2^/m^2^ in F PMI: <5.9 cm^2^/m^2^ in M, <4.1 cm^2^/m^2^ in F SMA: <33.9 HU in M, <30.9 HU in F	-	-	There was no difference among groups, considering postoperative complications, reoperation rate, and ICU admissions. Sarcopenic obesity was the only factor of overall postoperative morbidity (OR 1.241, *p* = 0.041).
Ryu et al. [48]	548	PDAC	PD, PPPD, PRPD	SMI	<50.18 cm^2^/m^2^ in M, <38.63 cm^2^/m^2^ in F	VFA/SMI	>2.5	Sarcopenic patients, compared to patients with normal muscle mass, presented a higher, but not significant percentage of major complications (Clavien–Dindo classification ≥ 3) (22% vs. 16.3%, *p* = 0.093), DGE (7.4% vs. 3.6%, *p* = 0.051), post-PD hemorrhage (3.4% vs. 3.2%, *p* = 0.894), wound infection (8.4% vs. 6.7%, *p* = 0.456), and bile leak (0.7% vs. 0%, *p* = 0.502). Five-year survival rate was also worse (23.4% vs. 28.4%, *p* = 0.046).
Aoki et al. [49]	180	PDAC	PD	SMI	<7 kg/m^2^ in M, <6 kg/m^2^ in F	-	-	Sarcopenia was identified as an independent risk factor for a lower recurrence-free rate (HR 4.48, 95%CI 1.68–11.98, *p* = 0.003) and poorer overall survival (HR 3.25, 95%CI 1.19–8.86, *p* = 0.021).
Pessia et al. [30]	76	PDAC	PD	SMI	≤52.4 cm^2^/m^2^ in M, ≤38.5 cm^2^/m^2^ in F	-	-	A higher duration of hospitalization (21 days vs. 17 days) and lower recurrence-free survival and OS were noted in sarcopenic patients compared to non-sarcopenic individuals.
Centonze et al. [50]	110	PDAC, DBDC, AC, IPMN, PNET, GIST, chronic pancreatis	PD	HUAC	<16.37 HU in M, <14.21 HU in F	-	-	The authors revealed a significantly higher rate of POPF type C in sarcopenic patients (50% vs. 11.4%, *p* = 0.005).
Sur et al. [51]	104	PDAC	PD, PPPD	TPAI	Mean TPAI < 5.8 cm^2^/m^2^	-	-	Low HU values were associated with serious surgical complications (r = −0.31, *p* = 0.0098) as well as the grade ≥ 3 of Clavien–Dindo complications (r = −0.29, *p* = 0.0183).
Takagi et al. [52]	291	PDAC, DBDC, AC, DC, IPMN	SSPPD	SBI	<68.5 cm^2^/m^2^ in M, <52.5 cm^2^/m^2^ in F	-	-	Higher rate of any infectious postoperative complications (67.3 vs. 40.2, *p* < 0.001), infected abdominal fluid collections (43% vs. 25%, *p* = 0.009), the incidence of bacteriemia (9.1% vs. 1.2%, *p* = 0.009), pneumonia (5.5% vs. 0.6%, *p* = 0.02), and intra-abdominal abscess (10.9% vs. 3.7%, *p* = 0.04) was observed in sarcopenic patients. The 5-year survival rates were lower compared to non-sarcopenic individuals (23.4% vs. 28.4%, *p* = 0.046).
Sui et al. [53]	150	PDAC, IPMN, PNET, AC, DC, DBDC	PPPD	SMI	<40.5 cm^2^/m^2^ in M, <33.5 cm^2^/m^2^ in F	-	-	A lower incidence of POPF type B and C was observed in sarcopenic patients (19.5% vs. 31.6%, *p* = 0.04). The 5-year overall survival in patients with sarcopenia was also lower (51.6 vs. 57, *p* = 0.009).
Nauheim et al. [54]	333	PDAC, IPMN, AC, DC, PNET, pancreatitis	PD	PMA	The lowest quartile in both sexes	-	-	Low PMA was independently associated with increased complication rates (OR 4.3, 95%CI 2.2–8.5, *p* < 0.01) and prolonged hospitalization (6.0 [5.0–8.0] days vs. 5.0 [5.0–6.0], *p* < 0.05).
Nishida et al. [55]	266	PDAC	SSPPD	SMI	<43 cm^2^/m^2^ in M with BMI <25 kg/m^2^ and <53 cm^2^/m^2^ in M with BMI of ≥25 kg/m^2^, <41 cm^2^/m^2^ in F	-	-	Patients with sarcopenia experienced significantly higher rates of major surgical complications stated as Clavien–Dindo ≥ 3 (34.1% vs. 15.7%, *p* = 0.001) as well as POFP type B and C (22.0% vs. 10.4%, *p* = 0.011).
Sandini et al. [56]	124	PDAC	PD, PPPD	TAMA index	<43 cm^2^/m^2^ in M with BMI <25 kg/m^2^ and <53 cm^2^/m^2^ in M with BMI ≥25 kg/m^2^, <41 cm^2^/m^2^ in F	-	-	The occurrence of abscesses was higher in sarcopenia (43.3 vs. 20.6, *p* = 0.007). The mortality rate was not significantly different between the groups (6.9% vs. 4.3%, *p* = 0.564).
Yoo et al. [57]	257	PDAC	PD	SMI	<49.5 cm^2^/m^2^	-	-	Low skeletal muscle was associated with lower 1-, 3-, and 5-year OS rates [86.2% vs. 71.9%, 39.7% vs. 30.3%, and 28.8% vs. 16.3 %, respectively (*p* = 0.026)].
Cai et al. [58]	129	PDAC, DBDC, AC, pancreatic cysticneoplasm, PNET, and chronicpancreatitis	PD, PPPD	SMI	<42.2 cm^2^/m^2^ in M, <33.9 cm^2^/m^2^ in F	-	-	Sarcopenia was a sensitive predictive marker of the occurrence of both POPF type B and C (OR 2.999, 95%CI 1.365–6.596, *p* = 0.006) and higher CCI scores above 26.2 (OR 3.425, 95%CI 1.541–7.613, *p* = 0.003).
Menozzi et al. [59]	103	PDAC	PD	SMI	<52.4 cm^2^/m^2^ in M, <38.5 cm^2^/m^2^ in F	-	-	There was a relationship between sarcopenia and digestive tract hemorrhages (OR 0.10, 95%CI 0.01 0.72, *p* = 0.03).
Xu et al. [60]	152	PDAC, other malignant and benign entities	PD	TPAI	≤4.78 cm^2^/m^2^ in M, ≤3.46 cm^2^/m^2^ in F	-	-	Sarcopenic patients presented a higher occurrence of major complications (40.7% vs. 7.5%, *p* = 0.000) and longer duration of hospitalization, but a higher percentage of in-hospital mortality was not significant (6.8% vs. 1.1%, *p* = 0.055).
Guarneri et al. [61]	371	PDAC	PPPD	SMI and SMA	SMI: <43 cm^2^/m^2^ in M with BMI <25 kg/m^2^ and <53 cm^2^/m^2^ in M with BMI ≥ 25 kg/m^2^, <41 cm^2^/m^2^ in F SMA: <41 HU in both sexes with BMI < 25 kg/m^2^ and <33 HU in both sexes with BMI ≥ 25 kg/m^2^	VFA/TAMA	>3.2	Sarcopenic obesity was associated with major postoperative complications (OR 1.675, 95%CI 1.02–2.98, *p* = 0.048), including POPF type A, B, and C (16.9% vs. 28.5%, *p* = 0.010).
Hayashi et al. [62]	169	PDAC, DBDC, IPMN, neuroendocrine tumor, AC	PD	SMI	<52.4 cm^2^/m^2^ in M, <38.9 cm^2^/m^2^ in F	VFA/SMI	>1.58	Sarcopenic obesity was a risk factor for developing POPF type B and C (OR 2.94, 95%CI 1.09–7.94, *p* = 0.033).

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
