# Peer review of "The Association Between Preoperative Sarcopenia and Sarcopenic Obesity and the Occurrence of Postoperative Complications in Patients Undergoing Pancreaticoduodenectomy for Periampullary Malignancies—A Literature Review"

_nutrients, 2024, doi:10.3390/nu16203569_

Round 1
Reviewer 1 Report
Comments and Suggestions for Authors
The manuscript titled "The association between preoperative sarcopenia and sarcopenic obesity and the occurrence of postoperative complications in patients undergoing pancreaticoduodenectomy for periampullary malignancies" by Jakub Ciesielka et al., explore the association between sarcopenia, sarcopenic obesity, and postoperative complications in patients undergoing pancreaticoduodenectomy (PD)
Major Comments:
- While the manuscript provides a clear aim of evaluating the association between sarcopenia, sarcopenic obesity, and postoperative complications in patients undergoing pancreaticoduodenectomy (PD), it would benefit from a more explicit hypothesis statement, strengthening the framing around the clinical relevance of these findings (i.e., how they could potentially alter preoperative care or risk stratification)
- The introduction gives a good overview of sarcopenia and sarcopenic obesity, but more recent studies on sarcopenia's impact on various cancers, including pancreatic cancer, could provide a stronger basis for the research In addition to the current literature on malnutrition and cancer-induced cachexia, it would be valuable to include recent studies emphasizing the inflammatory component of sarcopenia. Inflammatory processes, driven by cytokines such as IL-6, TNF-α, and IL-1, play a crucial role in the development of sarcopenia, particularly in patients with gastrointestinal cancers like pancreatic cancer. This is not solely a result of caloric deficiency or muscle loss due to cachexia but also due to systemic inflammation contributing to muscle degradation and impaired regeneration. Integrating these inflammatory mechanisms would strengthen the manuscript’s rationale and provide a more robust context for understanding sarcopenia’s multifactorial nature. For example, studies such as [insert PMID PMID: 39232731 and 36933563] have demonstrated the impact of chronic inflammation on muscle wasting, independent of nutritional status.
- The selection of studies for the systematic review appears thorough. However, the exclusion criteria, particularly in regards to different surgical approaches like total pancreatectomy, are not fully justified. Why was it important to limit the analysis to pancreaticoduodenectomy, and what might be the implications of excluding other procedures?
- The data regarding the association of sarcopenia and postoperative complications, such as pancreatic fistula and delayed gastric emptying, are compelling. However, the results section could benefit from a more structured presentation of key findings, especially with more robust statistical interpretation of p-values, odds ratios (OR), and confidence intervals (CI). For instance, highlight how sarcopenia was a predictor of specific complications in a concise tabular format to make it easier for readers to follow the findings.
5. The discussion section should address the potential mechanistic pathways linking sarcopenia to worse surgical outcomes in more detail. While the manuscript mentions inflammatory cytokines and muscle depletion, it could further explore how these processes directly impact recovery and postoperative complication rates. In the Discussion, when addressing the mechanisms behind sarcopenia and its role in cancer-induced cachexia, you could further strengthen the argument by citing PMID 39232731 and 37862577. Only such an example “The multifactorial nature of cancer-induced sarcopenia, particularly involving inflammatory cytokines like IL-6 and TNF-α (), has been demonstrated in recent studies, emphasizing the need for early identification and intervention.”
- A stronger critical analysis of conflicting results among studies, particularly those where sarcopenia did not predict complications, would also enhance this section.
- The conclusion reiterates the association between sarcopenia, sarcopenic obesity, and postoperative outcomes, but it could be more impactful by emphasizing the clinical implications of the findings. Specifically, how might these results change clinical practice regarding preoperative nutritional or physical prehabilitation? see PMID: 39275303 and Again PMID 37862577
- Recent studies have indicated that the use of pancrealipase in patients with exocrine pancreatic insufficiency leads to improved nutritional markers and weight gain, both of which are critical in combating sarcopenia (PMID: 37862577). Although direct evidence on the effect of pancrealipase on sarcopenia is limited, this could be discussed
Minor Comments:
- A few grammatical errors and awkward phrasing were noted, particularly in the results section. For example, the phrase "Sarcopenic obesity was also significantly adherent to some surgical complications" is unclear and could be reworded for clarity.
- The flowchart in Figure 2 depicting the study selection process is helpful, but the figure legend could be expanded to clarify the reasoning behind the exclusion criteria.
The quality of the English language in the manuscript is generally good, but there are some areas where it could be improved for better clarity and readability.
Author Response
Dear Reviewer,
We express great gratitude for your diligent correcting of the manuscript. In accordance with your
directions, we entered improvements to the following corrections:
Remark 1. While the manuscript provides a clear aim of evaluating the association between sarcopenia, sarcopenic obesity, and postoperative complications in patients undergoing pancreaticoduodenectomy (PD), it would benefit from a more explicit hypothesis statement, strengthening the framing around the clinical relevance of these findings (i.e., how they could potentially alter preoperative care or risk stratification)
Answer 1. We have elaborated more on the hypothesis to be more thorough.
“The aim of this review is to assess the impact of sarcopenia and sarcopenic obesity on postoperative outcomes in patients with periampullary tumors who underwent PD. This paper will discuss if there is a correlation between sarcopenia or sarcopenic obesity and postoperative complications in patients undergoing PD for PDAC.”
Remark 2. The introduction gives a good overview of sarcopenia and sarcopenic obesity, but more recent studies on sarcopenia's impact on various cancers, including pancreatic cancer, could provide a stronger basis for the research In addition to the current literature on malnutrition and cancer-induced cachexia, it would be valuable to include recent studies emphasizing the inflammatory component of sarcopenia. Inflammatory processes, driven by cytokines such as IL-6, TNF-α, and IL-1, play a crucial role in the development of sarcopenia, particularly in patients with gastrointestinal cancers like pancreatic cancer. This is not solely a result of caloric deficiency or muscle loss due to cachexia but also due to systemic inflammation contributing to muscle degradation and impaired regeneration. Integrating these inflammatory mechanisms would strengthen the manuscript’s rationale and provide a more robust context for understanding sarcopenia’s multifactorial nature. For example, studies such as [insert PMID PMID: 39232731 and 36933563] have demonstrated the impact of chronic inflammation on muscle wasting, independent of nutritional status.
Answer 2. We emphasized the inflammatory component of sarcopenia which mainly driven by cytokines previously. We added an integration that sarcopenia is a multifactorial reflection of a lethal disease with various mechanisms including the systemic, extensive inflammation mediated connected to the tumor microenvironment, immunological response, reduced food intake (as the consequence of systemic inflammation) and the relationship between the tumor and surrounding tissues.
“It is worth noting that sarcopenia in PDAC-burdened patients, is not only the result of decreased food intake, during the disease, but a multifactorial reflection of a lethal disease. The mechanisms leading to sarcopenia in patients with PDAC involve the ex-tensive systemic inflammation connected with the tumor microenvironment, immunological response, and the relationship between the tumor and surrounding tissues [14].”
Remark 3. The selection of studies for the systematic review appears thorough. However, the exclusion criteria, particularly in regards to different surgical approaches like total pancreatectomy, are not fully justified. Why was it important to limit the analysis to pancreaticoduodenectomy, and what might be the implications of excluding other procedures?
Answer 3. Pancreatoduodenectomy, differs in complexity, duration, indications and the possibility of developing certain complications compared to other types of pancreatic resections. Therefore, to ensure the homogeneity of the comparable groups and to allow appropriate inference, we limited ourselves to pancreatoduodenectomy only, excluding other types of pancreatic resections from the analysis.
“Case reports and reviews were excluded as only original papers were meant to be included. Studies were regarded as eligible if they included patients with confirmed periampullary malignancies and the co-occurrence of sarcopenia or sarcopenic obesity who were treated by PD. Papers which described other surgical methods were not included to remain consistency.”
Remar 4. The data regarding the association of sarcopenia and postoperative complications, such as pancreatic fistula and delayed gastric emptying, are compelling. However, the results section could benefit from a more structured presentation of key findings, especially with more robust statistical interpretation of p-values, odds ratios (OR), and confidence intervals (CI). For instance, highlight how sarcopenia was a predictor of specific complications in a concise tabular format to make it easier for readers to follow the findings
Answer 4. We have tried to present the results in different ways and remaining a single table with all data seemed as the best option as papers that were included in the study described different complications. It would be difficult to present them in another table and preparing separate tables for each complication would decrease readability of the manuscript drastically.
Remark 5. The discussion section should address the potential mechanistic pathways linking sarcopenia to worse surgical outcomes in more detail. While the manuscript mentions inflammatory cytokines and muscle depletion, it could further explore how these processes directly impact recovery and postoperative complication rates. In the Discussion, when addressing the mechanisms behind sarcopenia and its role in cancer-induced cachexia, you could further strengthen the argument by citing PMID 39232731 and 37862577. Only such an example “The multifactorial nature of cancer-induced sarcopenia, particularly involving inflammatory cytokines like IL-6 and TNF-α (), has been demonstrated in recent studies, emphasizing the need for early identification and intervention.”
Answer 5. In the section ‘Future directions’ we added a potential direction, based on a serum cytokine levels that may contribute to better postoperative care. However, the usefulness of such interventions requires further research.
“In the future, given the multifactorial nature of sarcopenia, one may wonder about the usefulness of determining pro-inflammatory cytokines in predicting the development of postoperative complications in patients undergoing PD, which could contribute to better perioperative care. However, the usefulness of such interventions requires further research.”
Remark 6. A stronger critical analysis of conflicting results among studies, particularly those where sarcopenia did not predict complications, would also enhance this section.
Answer 6. We elaborated more on the discussion. We believe that confronting studies that did not show association enhanced this part of the paper.
“Interestingly, in 8 studies POPF occurred more frequently in the patients with sarcope-nia or sarcopenic obesity but the difference was not significant. This is a controversial finding which was not sufficiently explained by most papers. More studies are needed to assess the correlation correctly.”
“However, most papers did not observe this correlation.”
“This was the only study that assessed this parameter.”
Remark 7. The conclusion reiterates the association between sarcopenia, sarcopenic obesity, and postoperative outcomes, but it could be more impactful by emphasizing the clinical implications of the findings. Specifically, how might these results change clinical practice regarding preoperative nutritional or physical prehabilitation? see PMID: 39275303 and Again PMID 37862577
Answer 7. We added the possible advantages of prehabilitation. However, there is no conclusion among the sources if prehabilitation enhance the outcomes. We focused mostly on papers regarding PDAC.
“Nutritional prehabilitation was introduced as a method that can enhance the outcomes of patients with pancreatic cancer. It can be beneficial for the postsurgical quality of life, nutritional status, and performance in sarcopenia-prone patients [63].”
Remark 8. Recent studies have indicated that the use of pancrealipase in patients with exocrine pancreatic insufficiency leads to improved nutritional markers and weight gain, both of which are critical in combating sarcopenia (PMID: 37862577). Although direct evidence on the effect of pancrealipase on sarcopenia is limited, this could be discussed
Answer 8. We added the idea of administering pancrelipase. We truly think that this will provide a better insight into how to counter sarcopenia and its complications.
“Another interesting possibility is administration of pancrelipase. Matsumoto et al. [66] demonstrated that in sarcopenic patients it helps increase the number of cycles of adjuvant chemotherapy (6 vs 3, p=0.03). The completion rate was also significantly greater when the enzyme was administered (86% vs 25%, p=0.007) [66]. This method can contribute to a better prognosis for sarcopenic patients with pancreatic cancer.”
We also fixed minor mistakes that we noticed.
We hope that our changes will meet Your expectations. We truly believe that those remarks were indeed important and our paper can benefit from them. We are grateful for them.
Best regards,
The Authors

Reviewer 2 Report
Comments and Suggestions for Authors
Comments attached in the word file below

Author Response
Dear Reviewer,
We express great gratitude for your diligent correcting of the manuscript. In accordance with your
directions, we entered improvements to the following issues:
Remark 1. Could be interesting to add some pathology in which this process is similar, like stroke for instance. It can bring a bit of perspective on this physiological mechanism in particular
Answer 1. Although, stroke’s mechanism is similar, we intended to focus omit acute diseases as sarcopenia is a chronic state that develops over long time.
Remak 2. Is there a connection with age too? In that case I believe that it can be good to include it, but concerning the pathology, I don´t know if it´s relevant to the study
Answer 2. There were no studies that associated TPAI with age, thus, we did not include this parameter.
Remark 3. I suggest that the affirmation that” CT is crucial” would be better if explained according to the facts exposed before. Maybe just highlighting the inaccuracy of the other methods.
Answer 3. We provided a summary describing the disadvantages of the other methods. It emphasizes the role of CT in the diagnostic process.
“Other methods are not as cost-effective or may provide false results depending on the hydration level. Moreover, CT is the only routinely preformed imaging method.”
Remark 4. : I recommend a better explanation for this consensus
Answer 4. In the current version, we stated that the consensus was reached between the authors.
“The discrepancies were solved by consensus between the authors.”
Regarding implementing the additional table in the discussion, thank You for Your remark. We are grateful for the guide to improve our article. On the other hand, we think that those information are mentioned in the tables already provided. We also believe that the manuscript contains large tables and adding another one may confuse readers or even make the paper difficult to process.
We added a brief idea of prehabilitation in the introduction. However, our paper does not focus on this topic. Our goal for mentioning prehabilitation was only to widen the readers’ perspective and show alternatives that should be assessed in the future.
“Prehabilitation in the patients with PDAC is was believed to have a positive impact on the outcomes, however, there is a lack of the multicenter studies that would support this view.”
We hope that our changes will meet Your expectations. We truly believe that those remarks were indeed important and our paper can benefit from them. We are grateful for them.
Best regards,
The Authors

Reviewer 3 Report
Comments and Suggestions for Authors
The aim of the study is to assess the impact of sarcopenia and sarcopenic obesity on postoperative outcomes in patients undergoing pancreaticoduodenectomy for periampullary tumors. This review seeks to clarify the relationship between these conditions and the incidence of postoperative complications, as existing literature presents varying results on this association
There are minor revisions to apply following my suggestion:
Abstract: Line 12-28: Improve clarity by separating the Background, Methods, Results, and Conclusions into distinct sections for better readability.
Aim of the Study: Line 46-48: Ensure that the aim is clearly articulated and consider rephrasing for conciseness.
Definition Consistency: Line 49-60: Clarify the definitions of sarcopenia and sarcopenic obesity, ensuring consistency with current guidelines and literature.
Methodology Detail: Line 176-181: Provide more specifics about the inclusion and exclusion criteria for study selection to enhance transparency.
Statistical Reporting: Line 19-23: Ensure consistent formatting when reporting odds ratios (ORs) and clarify significance levels for all statistics.
Discussion Depth: Line 27-28: Expand the discussion on clinical implications and future research directions based on findings.
Author Response
Dear Reviewer,
We express great gratitude for your diligent correcting of the manuscript. In accordance with your directions, we entered improvements to the following issues:
Remark 1. Line 12-28: Improve clarity by separating the Background, Methods, Results, and Conclusions into distinct sections for better readability.
Answer 1. We have implemented the remark. The abstract is now divided into sections more clearly.
“Abstract: Background: Sarcopenia and sarcopenic obesity, perceived as a reflection of cancer-induced cachexia, are often diagnosed in patients with periampullary malignancies. The pathophysiology of those conditions is multifactorial regarding the tumor microenvironment, immunological response, and the relationship to surrounding tissues.
Methods: The PubMed and SCOPUS databases were systematically searched between November 2023 and December 2023. 254 studies were primarily identified. Regarding the inclusion and exclusion criteria, 26 studies were finally included in the review.
Results: Evaluated papers disclosed that sarcopenia was significantly associated with a higher incidence of postoperative complications, including pancreatic fistula (POPF) type B and C, with the odds ratio (OR) ranging from 2.65 (95%CI 1.43–4.93, p=0.002) to 4.30 (95%CI 1.15–16.01, p<0.03). Sarcopenic patients also suffered more often from delayed gastric emptying (DGE) with an OR of 6.04 (95%CI 1.13–32.32, p=0.036). Infectious complications, postoperative hemorrhage, and intraabdominal abscesses occurred more often in sarcopenic patients. Surgical complications were also noted more frequently when sarcopenic obesity was present. Preoperative nutritional prehabilitation seems to reduce the risk of postoperative complications. However, more prospective studies are needed.
Conclusions: Sarcopenia and sarcopenic obesity were associated with a higher incidence of multiple postoperative complications, including POPF (type B and C), DGE, hemorrhage, and infectious complications.”
Remark 2. Line 46-48: Ensure that the aim is clearly articulated and consider rephrasing for conciseness.
Answer 2. We rephrased the aim. We hope that it is now clearly articulated.
“The aim of this review is to assess the impact of sarcopenia and sarcopenic obesity on postoperative outcomes in patients with periampullary tumors who underwent PD. This paper will discuss if there is a correlation between sarcopenia or sarcopenic obe-sity and postoperative complications in patients undergoing PD for PDAC.”
Remark 3. Line 49-60: Clarify the definitions of sarcopenia and sarcopenic obesity, ensuring consistency with current guidelines and literature.
Answer 3. The definition of sarcopenia differs in the literature. We included several definitions to show that there is no strict consensus how this state should be defined. We see that those explanations of sarcopenia may be vague, however, in the literature provided, this is how it is presented. Unfortunately, in the published guidelines we did not find any clear cut off to standardize the sarcopenic state.
Remark 4. Line 176-181: Provide more specifics about the inclusion and exclusion criteria for study selection to enhance transparency.
Answer 4. We elaborated the exclusion and inclusion criteria to make them more transparent.
“Case reports and reviews were excluded as only original papers were meant to be in-cluded. Studies were regarded as eligible if they included patients with confirmed per-iampullary malignancies and the co-occurrence of sarcopenia or sarcopenic obesity who were treated by PD. Papers which described other surgical methods were not in-cluded to remain consistency. The sarcopenia and sarcopenic obesity had to be as-sessed with the CT, according to the standardized protocols described in the literature [29,31,39]. Studies in which authors assessed PD and other surgical approaches, in-cluding distal pancreatectomy or total pancreatectomy, were excluded. The discrepan-cies were solved by consensus between the authors.”
Remark 5. Line 19-23: Ensure consistent formatting when reporting odds ratios (ORs) and clarify significance levels for all statistics.
Answer 5. The formatting was corrected to remain consistency.
Remark 6. Line 27-28: Expand the discussion on clinical implications and future research directions based on findings.
Answer 6. In the discussion, we stated that most papers noted no differences between the groups, thus, it is difficult to state clinical implications that are relevant.
We hope that our changes will meet Your expectations. We truly believe that those remarks were indeed important and our paper can benefit from them. We are grateful for them.
Best regards,
The Authors

Round 2
Reviewer 1 Report
Comments and Suggestions for Authors
The manuscript is generally improved after revision with a more refined hypothesis.
Comments on the Quality of English LanguageThe quality of the English language in the manuscript is generally good, but there are some areas where improvements can be made for clarity and readability. Some sentences are long and complex, making them harder to follow. In some places, there are missing commas or other punctuation marks that would help with the flow of the text.